



# Total solar irradiance using a traceable solar spectroradiometer

Dhrona Jaine[1], Julian Gröbner[1], Wolfgang Finsterle[1]

[1]Physikalisch-Meteorologisches Observatorium Davos, World Radiation Center (PMOD/WRC), Davos Dorf, Switzerland

*Correspondence to*: Dhrona Jaine (dhrona.jaine@pmodwrc.ch)

**Abstract.** Accurate, precise and traceable measurements of total and spectral solar irradiance measurements are fundamental for solar energy applications, climate studies, and satellite validation. In this study, we assess the performance and the quality of the data from a commercially available, compact BTS Spectroradiometer system, by comparing its spectrally integrated total solar irradiance (TSI) values with an electric substitution cavity radiometer (PMO2), which is traceable to the World Radiometric Reference (WRR). The resulting ratio between BTS Spectroradiometer system and WRR-traceable TSI is 0.9975 with a standard deviation of 0.0050. Applying a correction factor of (-) 0.34% to PMO2, accounting for the known offset between WRR and the International system of Units (SI) results in a relative difference between the BTS Spectroradiometer system derived TSI and PMO2 of +0.09% with a standard deviation of 0.0050 demonstrating good consistency between BTS derived TSI and the cavity radiometer.

This comparison confirms the precision and accuracy of the BTS spectroradiometer system, and its capability to deliver SI traceable TSI from spectrally resolved solar irradiance measurements. Its spectral resolution enables accurate measurements of spectral solar irradiance, which are essential, not only for determining total solar irradiance but also for retrieving key atmospheric gases such as water vapor, ozone, and aerosols, establishing its relevance as a compact instrument for atmospheric and climate research.

## 1 Introduction

The Earth's climate system is fundamentally governed by the interaction of solar radiation with atmospheric constituents. Accurate assessment of this interaction requires not only measurement of the total solar irradiance but also detailed characterization of its spectral distribution (IPCC 2021, 2023).

Total solar irradiance quantifies the amount of solar radiation reaching earth's surface often measured in watts per square metre (W/m²). The spectral character of the solar variations is important to the Earth's climate, as atmospheric constituents like aerosols and various constituent gases in the atmosphere selectively absorb and scatter the solar radiation in a wavelength dependent manner (Liou, K. N. (2002)). Therefore, determining parameters such as aerosol optical depth (AOD) and concentration of these gases from ground-based measurements is essential for accurately quantifying these radiative effects and validating satellite observations or radiative transfer models (Holben, B. N., et al. (1998), IPCC (2021)).



Understanding the variability of the solar terrestrial interactions requires therefore continuous, precise and accurate
monitoring of the solar spectral radiation distribution. Enhanced solar irradiance data play a crucial role in refining climate
models, ultimately leading to more reliable predictions of future climate scenarios (Wild, M et al., 2015, Kopp, G et. Al,
2011). Careful assessment of the spectral bands and the interaction of the radiation will enhance the quantification of
influential atmospheric gases and aerosols, which are necessary for the further improvement in the climate modelling and
studies.

Recent advancements in TSI and solar spectral irradiance measurements have been made possible by improvements in
measurement precision, particularly through space borne missions. Notably, the VIRGO (Variability of solar Irradiance and
Gravity Oscillations) instrument aboard the Solar and Heliospheric Observatory (SOHO), operated by PMOD/WRC, has
provided  the longest continuous space-based TSI records since 1996 (Fröhlich et al., 2006). Furthermore, the contributions
from the missions of the Laboratory for Atmospheric and Space Physics (LASP), such as the Solar Radiation and Climate
Experiment (SORCE) (Rottman et al., 2005), the Clouds and the Earth's Radiant Energy System (CERES) (Wielicki et al.,
1996) and the Total and Spectral Solar Irradiance Sensor (TSIS) (Coddington et al., 2021) provides essential global datasets
at the top of the atmosphere. However, estimating solar irradiance at the Earth's surface from such space-based
measurements involves significant relative uncertainties due to atmospheric influences from aerosols, clouds, and different
gases constituents (Wild, M et al., 2009). This highlights the necessity of high-quality ground-based measurements to better
quantify the interaction between the incoming solar radiation and the earth atmosphere.

In parallel, advancements in ground based spectroradiometer technology have significantly enhanced our capacity to capture
high-resolution and accurate solar spectral irradiance data across a broad wavelength range (Gröbner et al., 2019, Kouremeti
et al., 2022, Hülsen et al., 2022, Pereira et al., 2018, Egli et al., 2022). These advancements driven not only by technological
improvements but also by enhanced measurement methods and calibration techniques collectively contribute towards a more
comprehensive understanding of the Earth's radiation budget and solar–climate interactions.

Among these innovations, the Bi-Tec Sensor (BTS) spectroradiometer system (Zuber et al., 2018, Gröbner et al., 2023)
stands out as a commercially available, compact, and cost-effective system consisting of up to three array spectroradiometers
capable of delivering solar spectral irradiance measurements extending from 280 nm to 2150 nm.  Its portability and ease of
use make it an attractive instrument for deployment at ground-based monitoring stations.

The world radiometric reference (WRR) is the international standard for solar irradiance measurements, established by the
World Meteorological Organization (WMO) in 1979. It is realized by the World Standard Group (WSG), a set of six
absolute cavity radiometers at the Physikalisch Meteorologisches Observatorium Davos, World Radiation Centre
(PMOD/WRC). On the other hand, SI-traceable optical power is realized through cryogenic radiometers at National
Metrology Institutes like the National Physical Laboratory (NPL), providing traceability to SI units with uncertainties as low
as 0.02% (Fox et al. 1989). (Fehlmann et al. 2012) report that the WRR agrees well with the SI radiometric scale when
calibrated in power mode, but when operating in irradiance mode, WRR readings are systematically about 0.34 % higher



than SI-traceable values, primarily due to underestimated stray light contributions in the cavity radiometer, prompting efforts to align the WRR scale to the SI.

This paper introduces the use of BTS spectroradiometers to measure TSI, detailing its validation against the World Radiometric Reference (WRR) and its traceability to the International System of Units. The subsequent sections describe the instrumentation and methodology, followed by the presentation of results, a discussion of the findings, and concluding remarks.

## 2 Instruments

All the measurements were obtained at the PMOD/WRC located in Davos, Switzerland (46.803° N, 9.836° E) from January

2024 to March 2025 under clear sky conditions. It is located at an altitude level of 1590 meters above sea level which provides a stable and relatively clean environment for solar radiation measurements. PMOD/WRC hosts the World Radiation Center (WRC) on behalf of the World Meteorological Organisation. In particular, the Solar Radiometry Section of the WRC maintains and operates the World Standard Group of solar pyrheliometers which forms the Worldwide reference for Total Solar irradiance (World Radiometric Reference, WRR). The spectroradiometer system used in the study are two BTS

Spectroradiometers and the cavity radiometer PMO2, as the details will be explained in the following sections.

### 2.1 BTS Spectroradiometers

The Bi-Tec Sensor (BTS) spectroradiometers are the commercially available spectroradiometers developed by the company Gigahertz Optik GmbH. The system composed of two array-spectroradiometers based on a crossed Czerny-Turner design (Shafer et al. 1964), with an active stray light reduction with the help of optical bandpass and edge filters (Zuber et al. 2018)

covers a spectral range of 280 nm to 2150 nm in total. The spectral range from 280 nm to 1050 nm is covered by a 2048-pixel Si BTS2048-VL-TEC-WP spectrometer, which has a nominal spectral resolution full width at half maximum (FWHM) of 2 nm (Zuber et al. 2018). The spectral range from 950 nm to 2150 nm is measured using a BTS2048-IR-WP spectrometer, equipped with a 512-pixel extended InGaAs detector, offering a nominal spectral resolution of 8 nm (Gröbner et.al 2023). Each spectroradiometer has a collimator and is mounted on a solar tracker to measure only the direct solar spectral

irradiance.

### 2.1.1 BTS calibration and uncertainty budget for integrated solar irradiance spectrum

The responsivity is obtained by measurements of a transfer standard 1000 W tungsten-halogen lamp traceable to the SI and calibrated at the Physikalisch-Technische Bundesanstalt (PTB), Germany. The relative spectral uncertainty for direct spectral solar irradiance measurements were derived following the methodology outlined in Gröbner et al., 2023 and is shown in

Fig.1.




As can be seen in the fig.1, the relative uncertainty in the range 340 nm to 1750 nm is below 0.80 %. At shorter wavelengths the relative uncertainty increases significantly due to the low sensitivity of the BTS, while at wavelengths longer than 1750 (1750 to 2150 nm) the relative uncertainty is larger also of the order of 1.80% due to the noise of the extended InGaAs detector used in that spectral range.

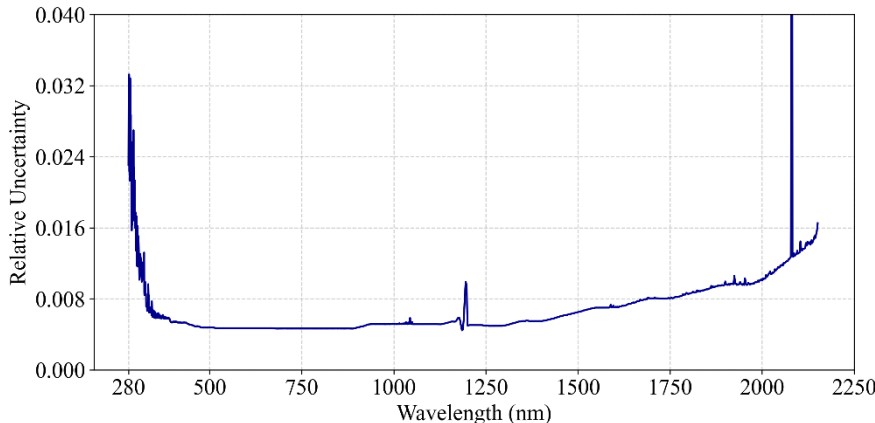

**Figure 1. Relative spectral uncertainty of direct spectral solar irradiance measurements presented as a function of wavelength. The localized relative uncertainty spike near 2080 nm is attributed to instrumental noise at the corresponding spectral pixel (hot pixel) and has been removed from the later analysis.**

To assess the overall relative uncertainty associated with the solar irradiance spectrum retrieved from the spectroradiometer, a total weighted mean uncertainty (JCGM, 2008) was calculated using the individual relative spectral uncertainties $u_\lambda$ across
all wavelengths. We followed a conservative approach and assumed that the relative spectral uncertainties of the reference lamp were fully correlated across the whole spectral range. The relative uncertainties at each wavelength were first weighted by the corresponding spectral irradiance:

$$w_\lambda = E_\lambda \cdot u_\lambda \qquad (1)$$

$E_\lambda$ is the spectral irradiance at wavelength $\lambda$ and $u_\lambda$ is the corresponding relative uncertainty.

The relative uncertainty of the total solar irradiance  u was obtained by numerically integrating the weighted uncertainties $w_\lambda$ over the whole spectral range:

$$u = \int_{\lambda 1}^{\lambda 2} E_\lambda \cdot u_\lambda \, d\lambda \qquad (2)$$

And dividing by the total spectral irradiance E ,

$$E = \int_{\lambda 1}^{\lambda 2} E_\lambda \, d\lambda \qquad (3)$$

The relative uncertainties u of the total solar irradiances obtained from the BTS spectroradiometer system therefore depend on the solar spectrum and were calculated for every solar spectrum of the BTS. The average relative uncertainties of the total




solar irradiances measured by the BTS Spectroradiometer system in the investigated period vary from 0.531 % to 0.534 %, giving an average relative uncertainty of 0.533 %.

## 2.2 Cavity Radiometer PMO2

The PMO-type instruments are absolute electrical substitution radiometers (Kendall et al., 1965) developed by Brusa and Fröhlich (1986), meaning they determine radiant power by substituting the absorbed optical power with an equivalent amount of electrical heating. This method relies on the principle that the temperature rise caused by absorbed radiation can be exactly matched by an equivalent electrical signal, allowing the radiant power to be quantified without the need for calibration against another radiometer. It will measure the solar radiant power received through an aperture with known area and hence is measuring the absolute value of the total solar irradiance. Such instruments form the backbone of high-precision solar irradiance measurements, including the World Radiometric Reference. The PMO2, one of the six instruments in the WRR ensemble, has an associated standard uncertainty of 0.03% with respect to WRR. These ambient temperature instruments measure direct normal irradiance with an expanded uncertainty of 0.19% with 95% confidence level with respect to international system of units (Fehlmann et. al 2012).

## 3 Methodology for finding total solar irradiance from the measurements.

To enable effective validation against cavity radiometers, which measures broadband solar irradiance across the full solar spectrum, it was necessary to extend the BTS spectroradiometer-derived spectrum beyond its measured spectral range of 2150 nm. This was achieved by using the radiative transfer model (RTM) *libradtran* (Mayer B et al., 2005) to model solar

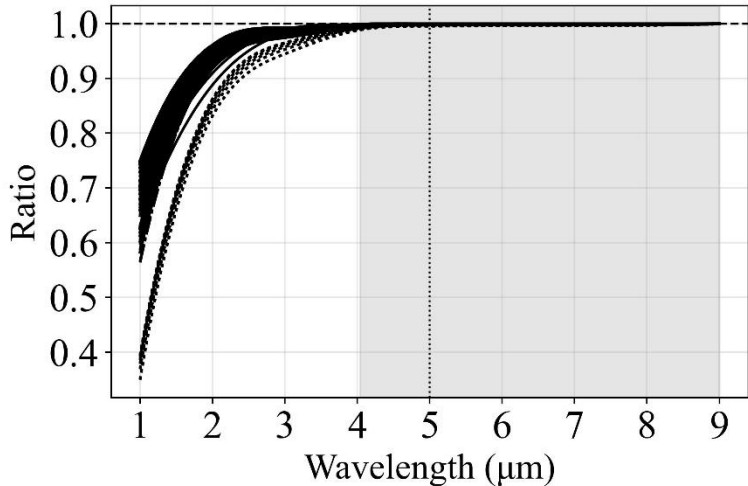

**Figure 2:** Ratio of spectral irradiance at each wavelength to the total spectral irradiance, illustrating the cumulative contribution of different wavelengths and different conditions typical of Davos. The grey shaded area represents the spectral interval contributing more than 90% of the total solar irradiance (TSI). The grey vertical dotted line marks the maximum upper limit of wavelength adopted for the study. The horizontal line at 1 represents the normalized maximum contribution of shortwave solar radiation.





spectra over an extended spectral range, aiming to capture the majority of shortwave component of the TSI. To determine this spectral range, the modelled spectra under typical atmospheric conditions of Davos were cumulatively integrated to identify the wavelength interval that accounts for at least 99.9% of the TSI. Fig. 2 illustrates the cumulative solar spectral irradiance, showing that the percentage of shortwave radiation captured by 5000 nm ranges from a minimum of 99.51% to a maximum of 99.97%, with 95% of the values being larger than 99.96%. These findings confirm that 5000 nm is a suitable

upper boundary for obtaining the TSI from the solar spectrum at Davos. Consequently, the spectral interval, 280 nm to 5000 nm was adopted for all further analysis. The BTS spectroradiometer system directly measures solar spectral irradiance in the wavelength range of 280 nm to 2150 nm. Since this range does not capture the full solar spectrum, the contribution from the remaining spectral interval (2150 nm to 5000 nm) is estimated using radiative transfer simulations. These simulations consider specific atmospheric conditions such as water vapor, aerosol properties, and solar zenith angle corresponding to

each solar measurement.

To extend the measured spectral range and estimate the TSI, a machine learning approach based on the XGBoost (XGB) algorithm (Chen et al., 2016) was developed. This model predicts the fractional contribution $\mathbf{R}$ of the 280 nm to 2150 nm range to the total TSI based on key atmospheric and optical parameters as a function of

$$\boldsymbol{R} = f(sza, wv, \alpha, \beta) \tag{4}$$

where R is defined as

$$R = \frac{\int_{2150}^{5000} I(\lambda)d\lambda}{\int_{280}^{5000} I(\lambda)d\lambda}. \tag{5}$$

with I representing the modelled solar irradiance spectrum. The total TSI is then reconstructed using the relation:

$$\boldsymbol{TSI} = \frac{\boldsymbol{TSI(BTS)}}{1 - \boldsymbol{R}} \tag{6}$$

where the numerator represents the integrated solar irradiance measured by the BTS, and the denominator accounts for the modelled fraction of the spectrum. This methodology allows the extension of the BTS measurement to full-spectrum TSI, enabling direct comparison and validation against broadband cavity radiometer data. Further details on the radiative transfer modelling and XGB implementation are provided in the following sections.

**3.1 Radiative Transfer model simulations**

The model simulations were performed for every measured solar spectrum in the period from January 2024 to March 2025. These measured solar spectra covered a variety of could-free atmospheric conditions found at Davos. Among the input parameters used in the model simulations, the spectral aerosol optical depth was directly retrieved from the measurements from BTS spectroradiometer, while the remaining parameters were obtained from ancillary measurements available at Davos, such as atmospheric pressure, integrated water vapour, and total column ozone. Actual values for carbon dioxide was

taken from the Copernicus Atmosphere Monitoring Service (CAMS). The reference top of atmosphere solar spectrum used





for the model calculations was the TSIS-1 HSRS solar spectrum (Coddington et al., 2021), which was validated against ground-based measurements (Gröbner et al., 2023).

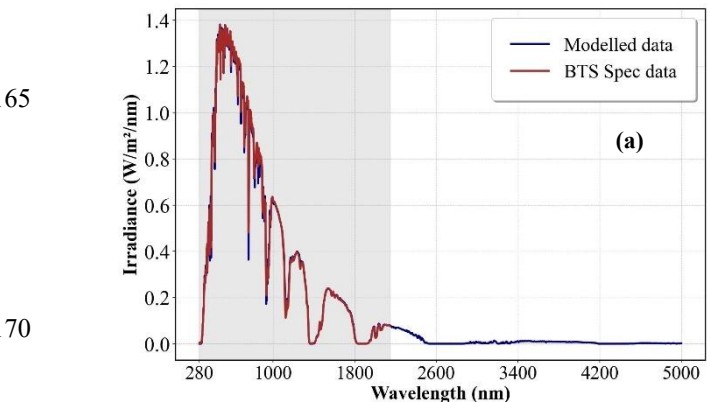
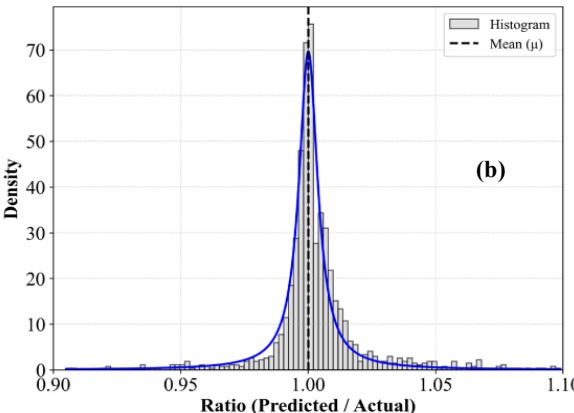

**Figure 3(a): Solar spectrum on 2023.09.11 at 12:31:06 CET. Blue curve: The solar spectral irradiance of the modelled spectrum from 280 nm to 5000 nm; red curve and shaded region: the measured solar spectrum from the BTS Spectroradiometer system. Figure 2(b): Ratio of the modelled to the measured solar irradiance spectra, based on 3442 spectra over the wavelength range of 280 nm to 2150 nm. A Voigt fit to the ratio distribution is shown in blue, illustrating the strong agreement between the modelled and measured spectra.**

The model solves the radiative transfer equation and simulates solar irradiance spectra by generating a synthetic atmosphere based on profiles of key atmospheric species. These profiles were normalized according to the corresponding atmospheric variables associated with each measurement, ensuring that the input conditions accurately reflected the observed state of the atmosphere at each given time, thereby calculating the full solar spectrum. This approach allowed to extend the spectral range of the BTS spectroradiometer beyond its initial upper limit of 2150 nm as shown in the figure 3(a). Furthermore, the accuracy of the modelled spectra relative to the BTS spectroradiometer measurements was assessed for 3442 could-free spectra by calculating and averaging each spectral ratio between these two datasets in the spectral range from 280 nm to 2150 nm. The distribution of ratios between the measurements and model was best described by a Voigt function-a combination of Gaussian and Lorentzian profiles, capturing both random and systematic deviations; this function was therefore fitted to the histogram of these ratios, shown in Figure 3(b). Resulting in a high correlation coefficient of 99.15%, with an average ratio of 1.003 and standard deviation of 0.0189 indicating excellent agreement between the modelled and observed spectral irradiance.

## 3.2 Sensitivity analysis

A sensitivity analysis was performed to investigate the relative importance of the input parameters of the radiative transfer model on the calculation of the extension of the measured solar spectrum by the BTS Spectroradiometer system. A





comprehensive sensitivity analysis was conducted using the One-At-a-Time (OAT) method to assess the individual influence

190    of key atmospheric parameters on the modelled

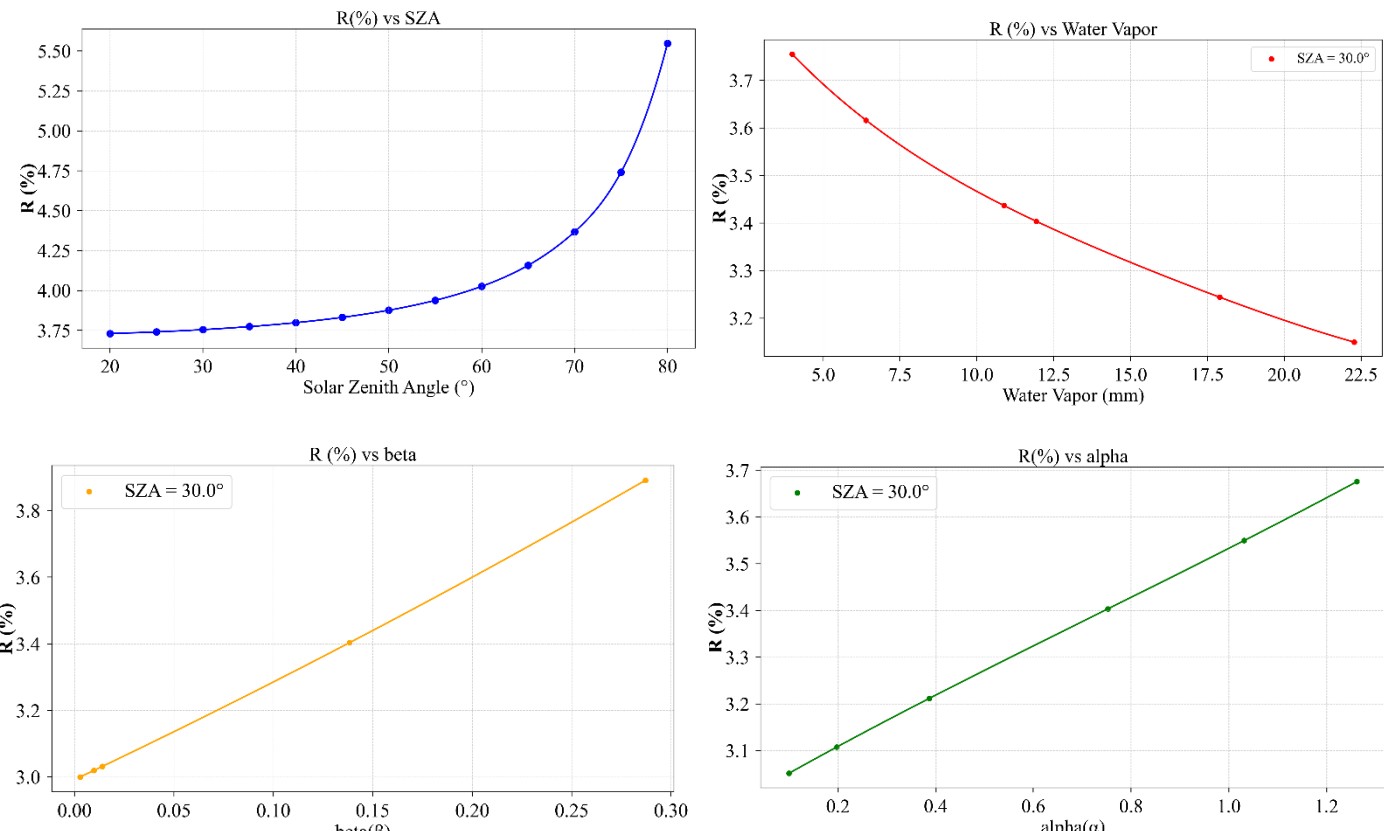

**Figure 4: Illustrating the results of sensitivity analysis showing the variation in R (see Eq. 4) in the extended spectral region from 2150 nm to 5000 nm in response to key atmospheric parameters. The baseline values representative of typical atmospheric conditions in Davos was used: aerosol angström parameters ($\alpha$) = 1.26, ($\beta$) = 0.13, and water vapour (WV) = 11.93 mm, carbon dioxide = 425 ppm, ozone = 325 DU.**

solar irradiance spectrum in the wavelength range of 2150 nm to 5000 nm. Parameters such as solar zenith angle, water vapor content, ozone column amount, angström parameters ($\alpha$ and $\beta$) were varied systematically within physically realistic bounds for the location of Davos, while all other inputs were held constant at baseline values. This approach allowed for the

195    isolation of each parameter's effect on the spectral range longer than 2150 nm of the solar spectrum. As illustrated in Fig.4, the solar zenith angle (SZA) and water vapor have a significant impact on the fractional contribution of solar irradiance in the 2150 nm to 5000 nm spectral region. Specifically, the variation in solar zenith angle from 20° to 80° results in an increase in the fractional contribution of this spectral region from 3.75% to 5.50%. This change accounts for the increased atmospheric path length at higher SZAs, which enhances the scattering and absorption by shorter wavelengths resulting in

200    the increase in the proportion of near infrared radiation. In addition, the contribution from the aerosol properties such as




angstrom exponent (alpha) and turbidity coefficient (beta) also notably affects the spectral contribution. Changes in alpha from 0.0 to 1.3 (typical range in Davos) showed an increase from 3.0% to 3.6% in R (while an increase in beta from 0.00 to 0.30 results in a shift from 3.0% to 3.9%. This effect is due to the influence of aerosols in scattering and absorption properties of the atmosphere, altering the spectral radiation distribution of the solar spectrum. Conversely, increasing water vapor from 4 mm to 22.5 mm leads to a decrease in the same spectral contribution from 3.8% to 3.2%. This reduction is attributed to the strong absorption characteristics of water vapor in the near-infrared range, particularly between 2150 to 5000 nm, which reduces the transmittance of radiation in this spectral band.

The sensitivity analysis was used for the subsequent development of the machine learning model by identifying the key atmospheric parameters governing the fractional contribution from 2150 nm to 5000 nm of the solar spectrum. Demonstrating that the SZA, aerosol properties and water vapour are the primary drivers. These findings emphasize the selection of input features for the machine learning approach, ensuring that it is both physically informed and optimized for predictive accuracy.

## 3.3 Machine Learning Approach (XGB regression model)

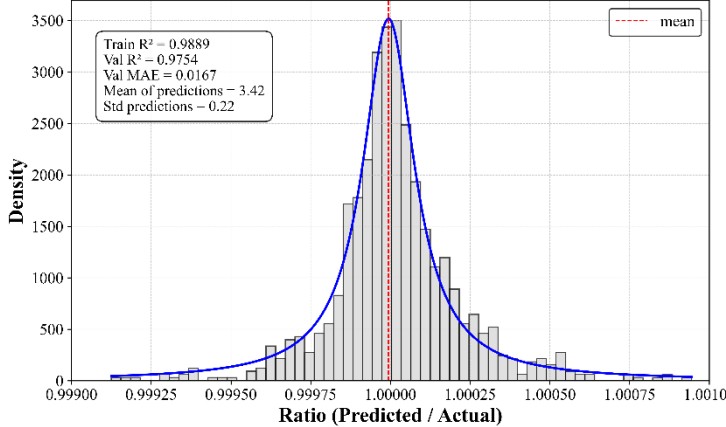

**Figure 5: Ratio of the predicted spectral extension R to the radiative transfer model–based spectral extension of solar irradiance over the wavelength range 2150 nm to 5000 nm. A Voigt fit to the ratio distribution is shown in blue, illustrating the strong agreement between the predicted and modelled spectra.**

However, due to the computational intensity of the model simulations, a faster and more efficient approach was needed for large-scale analysis and real-time calculations. A predictive machine learning model based on the Extreme Gradient Boosting (XGB) regression (Chen, T., et al. 2016) was implemented. It is a well-suited algorithm working on the principle of gradient boost decision trees for capturing complex, non-linear relationships. This model utilized the results of the sensitivity



analysis for the identification of the variables influencing the extended spectral region (2150 nm to 5000 nm). Multiple combinations of the four selected atmospheric variables were tested and evaluated to determine optimal configuration for model performance. The highest performance was achieved when the parameters angström coefficients (α and β), water vapor content, and solar zenith angle were used all together. The accuracy of the predicted spectral extension from the machine learning model, relative to that obtained from a radiative transfer model, was assessed using the ratio between the two datasets. A voigt function was fitted to the histogram of these ratios, yielding a high training correlation coefficient ($R^2$) of 0.9889 and a validation correlation coefficient of 0.9754, with a mean value of 0.999 and a standard deviation of 0.038%, demonstrating the excellent agreement and the accuracy of the machine learning model under varying atmospheric conditions.

### 3.4 Uncertainty of TSI

The relative uncertainty in the estimation of the total solar irradiance (TSI) from BTS solar irradiance measurements arises from two sources (a) BTS measurements as detailed in section 2.1.1 and (b) the relative uncertainty in the extended spectral region, which is evaluated based on the variability of key input parameters within the machine learning model as well as the uncertainty of the machine learning model itself. To assess the contribution of individual input parameters to the relative uncertainty of the model in the calculation of R (see Eq.4 and Eq.5), a sensitivity analysis was conducted as shown in Fig. 4. The relative uncertainty contributions of each parameter to the TSI was estimated as follows: the relative uncertainty of SZA was estimated at 0.5° at 75°, which corresponds to a time uncertainty of about 30 seconds; the relative uncertainty in the GPS derived integrated water vapour was estimated at 1 mm as described in Nyeki et al. (2005), while the relative uncertainties of the aerosol parameters α and β were set to 0.1, and 0.01, respectively. The corresponding relative uncertainties in R were then retrieved from the curves shown in Fig. 4, where $u_{Rsza}, u_{Riwv}, u_{R\alpha}, u_{R\beta}$ are the relative uncertainties of R with respect to the parameters subject to the extension of the spectrum . Using the estimated relative uncertainties of the parameters, $u_{Rsza} = 0.054\%$, $u_{Riwv}= 0.030\%$, $u_{R\alpha} = 0.052\%$, and $u_{R\beta}= 0.031\%$, and the relative uncertainty of the machine learning model $u_{RML}=0.038\%$, the resulting combined relative uncertainty of R is then

$$u_R = \sqrt{u_{Rsza}^2 + u_{Riwv}^2 + u_{R\alpha}^2 + u_{R\beta}^2 + u_{ML}^2} = 0.0945\% \qquad (7)$$

This results in a combined relative standard uncertainty in the model extrapolation, R of 0.0945%. When combined with the BTS measurement uncertainty, collectively contributes to the uncertainty in the estimation of the TSI,

$$u_{TSI} = \sqrt{u_{BTS}^2 + u_R^2} \qquad (8)$$

This uncertainty can be estimated by following Eq.8 and it results in a relative expanded uncertainty of 1.081% (95% confidence, assuming a normal probability distribution).





## 4 Result and discussion

This spectral extension R of the solar spectral irradiance measurements using the machine leaning approach enabled a robust comparison with TSI measurements from the cavity radiometer (PMO2). The top panels of Figure 6 show the time series of TSI from both PMO2 and BTS spectroradiometer while the lower plots illustrate the ratio between the TSI of both instruments. The left panel presents the comparison of the BTS spectroradiometric system relative to the World Radiometric Reference (WRR), while the right panel shows the comparison after applying the WRR-to-SI offset to the

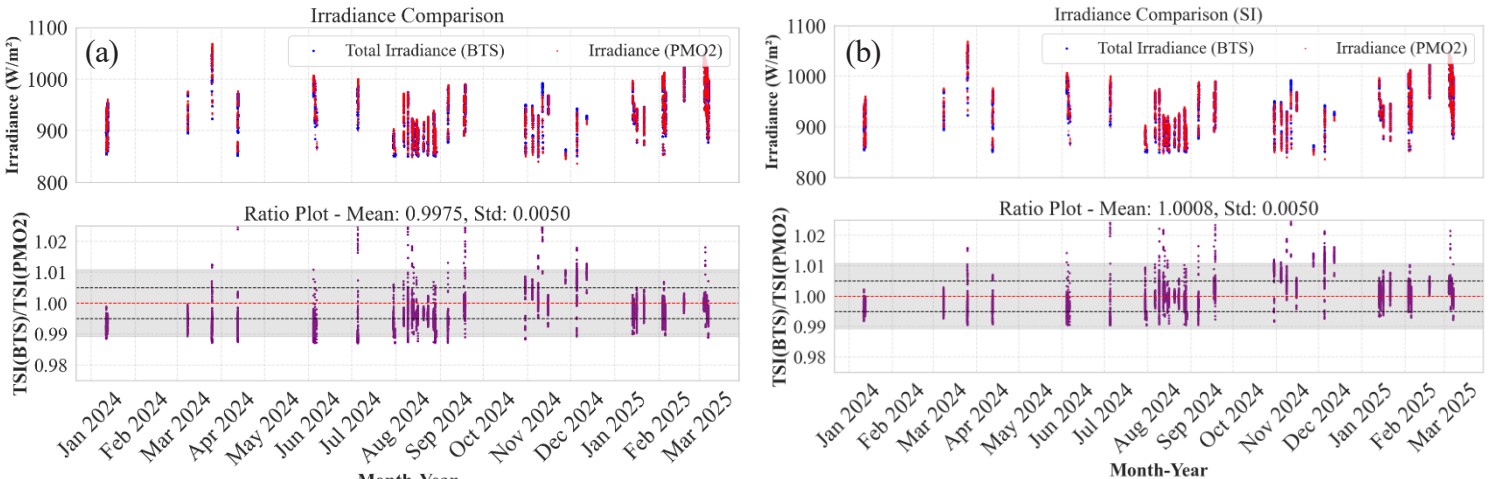

**Figure 6: Top panel shows the time series of total solar irradiance (TSI) measurements obtained from the BTS spectroradiometer and the cavity radiometer (PMO2). Bottom panels display the ratio between the TSI measurements from both instruments. The grey shaded region represents the combined uncertainty of the BTS spectroradiometer, the extended region of the spectrum (2150 to 5000nm) and PMO2 measurements, with the red line indicating the degree of agreement between the two instruments. The black dashed lines mark the ±5% bounds which is the standard deviation. Panel (a) illustrates validation against the World Radiometric Reference (WRR), while panel (b) demonstrates traceability to the International System of Units.**

PMO2 measurements. The statistical analysis using the Concordance Correlation Coefficient (CCC), the tool which accounts for both accuracy (closeness to y=x) and precision (Pearson r) of the instruments, indicates a value of 0.9688 (Lin, L. I. (1989)), reflecting a strong agreement between the instruments, as depicted in the top plots. The shaded region in the lower plots represents the combined expanded relative uncertainty of the instruments. As discussed previously, the TSI derived from the BTS

spectroradiometer system has an expanded relative uncertainty of 1.068% (k=2), where k=2 denotes a coverage factor that corresponds to a 95% confidence level, assuming a normal probability distribution. As already discussed in the section 3.4 the calculation of the extended region of the spectrum is associated with an expanded uncertainty of 0.189% (k=2). Similarly, the cavity radiometer PMO2 has an expanded relative uncertainty of 0.19% (k=2) with respect to SI. The combined expanded relative uncertainty is 1.098% (k=2). The ratio between the BTS and PMO2 using WRR traceability gives a mean value of 0.9975 with a standard deviation of 0.0050.





It is also important to note that, for SI traceability, the WRR carries a known offset of -0.34% (Fehlmann et. al 2012, IPC XII 2016) relative to the SI. When this correction is incorporated in the data of the PMO2 radiometer, the resulting ratio between the TSI values measured by the BTS Spectroradiometer and the reference cavity radiometer is 1.0008 with a standard deviation of 0.0050 (see Figure 6b). This demonstrates the robust agreement of BTS spectroradiometer with the World Radiometric Reference instrument PMO2 cavity radiometer, thereby confirming that the validation have been successfully met.

The result demonstrates that the BTS spectroradiometer system calibrated using SI-traceable reference standards in the laboratory, can provide integrated irradiance values that are consistent with the traditional broadband cavity radiometers, bridging the gap between spectrally resolved and broadband TSI measurements which is crucial for climate research. The demonstrated consistency from spectrally resolved data not only validates the BTS spectroradiometer system but it can also be used as a reliable and cost-effective instrument for retrieving both integrated and spectrally resolved solar irradiance. Its comparatively broad spectral range further enhances its utility by enabling the retrieval of atmospheric gases, making it well-suited for a wide range of remote sensing applications. Furthermore, it validates the capability of the BTS spectroradiometer system as a standard instrument for spectral solar irradiance retrievals, particularly in field conditions where traceability and reliability are important. In this context, the results contribute meaningfully to the broader goal of establishing a unified and SI traceable irradiance measurement system for spectral as well as total solar irradiance.

## 5 Conclusion

The study successfully demonstrated the validation and SI traceability of TSI derived from the BTS spectroradiometer system. By incorporating the known -0.34% WRR to SI offset into the PMO2 cavity radiometer data, we validated the TSI retrieved from the BTS spectroradiometer system, resulting in a good agreement between the BTS and cavity radiometer PMO2 with a mean ratio of 1.0008 and a standard deviation of 0.0050. This confirms that the BTS spectroradiometer system can reliably reproduce the broadband TSI values consistent with the WRR and SI traceable measurements.

Furthermore, this result has also broader implications: It confirms that a solid-state, spectrally resolved instrument like the BTS spectroradiometer system not only matches the performance of traditional broadband cavity radiometers but also provides enhanced functionality through spectral irradiance measurements. This capability positions the BTS spectroradiometer system as a versatile and SI traceable instrument for solar irradiance monitoring, radiative transfer validation, and potentially for the retrieval of the atmospheric gases and aerosols which are influencing the solar radiation reaching the earth surface. Its cost-effectiveness and large spectral coverage make it a valuable tool for long term climate monitoring, remote sensing, and atmospheric research. Overall, this work contributes a meaningful advancement toward a unified, traceable framework for solar radiation observation, benefiting the scientific community engaged in Earth system studies.



## Acknowledgment

D.J. was supported through the project Spectral Solar Irradiance Measurements using CompAct spectrometers
to retrieve ECVs (SIMCA), funded by the Swiss National Science Foundation under Grant Nb. IZCOZ0_220355.

## Data availability

The main datasets and necessary python scripts used in this study are available on Zenodo, https://zenodo.org and
specifically at https://doi.org/10.5281/zenodo.16910121.

## Author contribution

D.J analysed the BTS and PMO2 datasets and wrote the manuscript. J.G supported the analyses and the formulation of the
manuscript. W.F provided the PMO2 (WSG) dataset for the validation.

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
