# Peer review of "Total solar irradiance using a traceable solar spectroradiometer"

_EGUsphere, 2025_

## Author Comment (AC1)

**Reviewer 1 Comments:**

**Thank you so much for the comments.**

**In Section 2.1:**

**What is the FOV of the BTS spectroradiometer? Does it match the 5 deg FOV of the PM02?**

Ans: Thank you for the comment, which we did not consider in our manuscript. The BTS spectroradiometer has a field of view (FOV) of 2.4° and the PMO2 cavity radiometer have 5°. We quantified the FOV mismatch bias with the radiative transfer model SMARTS using the conditions of PMOD/WRC (including extreme cases) and found the DNI bias due to FOV differences to lie between approximately 0.03% to 0.11% of the total DNI in the extreme case aerosol conditions. This uncertainty contribution to the combined uncertainty of the PMO2 and BTS is well below our other uncertainties, so we regard the FOV effect as minimal for our conclusions. We have added a section on this topic in our manuscript at section 3.4 "Uncertainty of TSI" in page number 10, describing this effect.

The figure depicting the various aerosol conditions found at the measurement site during our study (PMOD/WRC)

O.06
E

O.06
E

O.06
E

O.04
O.06
O.06
O.07
O.08
O.09
O.09
O.00
O.00 -

**How often is the BTS calibrated? If only initially, how do you guarantee that it is stable?**

Ans: The BTS responsivity is monitored every 2 months using a 250 W tungsten halogen lamp (KS32). Once per year the BTS is also calibrated in the optical laboratory of PMOD/WRC using a transfer standard FEL 1000 W lamp traceable to the SI (calibration certificate obtained from the German Metrology Institute PTB). At the same time, KS32 is recalibrated as well.

**In Section 3:**

The caption for Fig. 2 is incorrect in that the grey area does not represent 90 % of the TSI. Addressed in the caption of the manuscript in section3.

Perhaps an insert that blows up the 4000 - 5000 nm region in Fig. 2 would clarify the points made in lines 133 and 134.

Addressed in the manuscript

I did not understand the necessity of a machine learning approach since one needs the model inputs (eqn. 4) to estimate the 2150 - 5000 nm contribution for machine learning or the model runs; why not just run the model to calculate the contribution?

Ans: Because repeated radiative transfer model simulations in libRadtran are computationally expensive. The spectral contribution in the 2150nm to 5000 nm region depends on several atmospheric inputs such as solar zenith angle, precipitable water vapour, and aerosol angstrom exponents (see Eq. 4), and a full DISORT-based forward model must be executed for each unique combination of these parameters. For long time series or real-time applications, this would require a very large number of model runs, resulting in increased computational cost and time.

Instead, a machine learning (ML) approach serves as an efficient replacement to the radiative transfer model for the calculation of TSI. Once trained on a representative set of libRadtran simulations, the model can reproduce the 2150 nm to5000 nm contribution with negligible computational effort. Thus, the ML framework enables fast predictions while still retaining the physical relationships embedded in the original forward simulations, making applications feasible.

**In Section 4:**

Fig. 6 is difficult to examine. Perhaps a blow up of just one vertical grouping would more clearly show the degree of agreement. I think you could eliminate the left part (a) of this figure. Addressed in the manuscript

Other:

Line 44 "gases constituents" "gases" Addressed in the manuscript

Look for "could" that should be changed to "cloud" in at least two places. Lines 156 and 178. Addressed in the manuscript in the line 156 and 178

In Fig. 2 caption "grey vertical" "vertical" Addressed in the manuscript

---

## Author Comment (AC2)

Reviewer comments #2

Thank you so much for the comments.

Specific comments:

Section 3. 1

Could you give more details on the input for the radiative transfer model calculations and on how you conducted the calculations (note: libRadtran is not a radiative transfer model but a library for radiative transfer, i.e. a collection of different solvers and band parametrization models (correct that in line 129)). Specifically:

Ans: We performed the simulations with libradtran, which is a library for radiative transfer which provides several independent solvers such as DISORT (Discrete Ordinate Solver), MYSTIC (Monte Carlo Solver) etc. and band parameterizations such as REPTRAN, LOWTRAN etc. We used DISORT as solver and the program "uvspec". The DISORT solver were being used with number of streams 16 for better accuracy. The specific parameters used for the model calculations are described in the manuscript at lines 156-165 (section 3.1).

- what kind of wavelength grid/band parametrization model did you use to be consistent with the spectra from the BTS? What about beyond 2150 nm? what solver did you use?

**Ans:** To ensure full consistency with the Bi-Tec Sensor (BTS) spectroradiometer, the model spectra were convolved with the BTS spectral slit functions, which are available every 5 nm in the range from 280 nm to 1000 nm, and every 25 nm in the range between 1000 nm and 2150 nm. The typical spectral resolution (defined at full width at half maximum), is 2 nm in the range 280 nm to 1050nm and 8 nm for the rest up to 5000 nm.

- line 173: "synthetic atmosphere". You may describe this a bit more detailed. You used the (US-) Standard atmosphere, normalized to the indicated observed atmospheric parameters, I guess?

**Ans:** (Addressed in the manuscript at lines 174-176) Yes, you are right. The term synthetic atmosphere indicates the US standard atmosphere (AFGLUS) which is then normalised using the observed atmospheric parameters such as Pressure, altitude, aerosol angstrom parameters, water vapour, ozone column, carbon dioxide at the site of PMOD/WRC Davos, Switzerland.

**Section 3.2**

- Have you also studied the impact of microphysical aerosol properties (e.g. single scattering albedo (SSA)) on the fractional contribution R as the aerosol type may also change the solar spectra substantially?

Ans: Direct normal solar irradiance is only affected by the atmospheric extinction (absorption and scattering), and not by the single scatter albedo SSA. However, a minor contribution from forward scattered radiation, entering the field of view, could have a SSA dependence. We evaluated the sensitivity of single scattering albedo (SSA) and found its influence on the direct normal irradiance (DNI) to be negligible, with ratio between the DNI at different SSA is 0.9998.

Because forward scattered radiation by aerosols and other atmospheric constituents, will contribute circumsolar radiation into the instrument field of view, we quantified a possible field of view (FOV) bias due to the different field of views of the BTS and the cavity radiometer PMO2. The BTS spectroradiometer (FOV of  $2.4^{\circ}$ ) and the PMO2 cavity radiometer (FOV 5°) were assessed using the radiative transfer model SMARTS over the aerosol conditions found at Davos ( $\alpha$ ,  $\beta$ ). The resulting DNI bias from the FOV mismatch is approximately 0.03 % – 0.11 % in those conditions. This uncertainty contribution to the combined uncertainty of the PMO2 and BTS is well below our other uncertainties, so we regard the FOV effect as minimal for our conclusions. We have added a section on this topic in our manuscript at section 3.4 "Uncertainty of TSI" in page number 10, describing this effect

(see also comment of reviewer 1).

In general: Define abbreviations once at their first occurrence and then use them throughout the manuscript (e.g.: Lines 55, 73: World Radiometric Reference is defined in line 9; line 73 World Standard Group is defined in line 56; line 77: the Bic-Tec Sensor (BTS) is defined in line 51 (or should be defined in line 7). Addressed in the manuscript

Line 31: Kopp et al. Addressed in the manuscript in the line 31

Line 144: May define sza, wv,  $\alpha$ ,  $\beta$  here (instead in section 3.2). Instead of water content, I would use precipitable water, pw, or integrated water vapor, iwv, as you used a column measure in the sensitivity analysis.  $\alpha$  and  $\beta$  is to my knowledge termed as Angstrom (or Ångström)  $\alpha$  and Angstrom (Ångström)  $\beta$ , respectively. Addressed in the manuscript

Line 189: May add a reference for the OAT method. Addressed in the manuscript in the line 189

Line 216: delete "However" Addressed in the manuscript in the line 216